# The Skin Histopathology of Pro- and Parabiotics in a Mouse Model of Atopic Dermatitis

**DOI:** 10.3390/nu16172903

**Published:** 2024-08-30

**Authors:** Hun Hwan Kim, Se Hyo Jeong, Min Yeong Park, Pritam Bhagwan Bhosale, Abuyaseer Abusaliya, Jeong Doo Heo, Hyun Wook Kim, Je Kyung Seong, Tae Yang Kim, Jeong Woo Park, Byeong Soo Kim, Gon Sup Kim

**Affiliations:** 1Research Institute of Life Science, College of Veterinary Medicine, Gyeongsang National University, Jinju 52828, Republic of Korea; shark159753@naver.com (H.H.K.); tpgy123@gmail.com (S.H.J.); lilie17@daum.net (M.Y.P.); shelake.pritam@gmail.com (P.B.B.); yaseerbiotech21@gmail.com (A.A.); 2Biological Resources Research Group, Gyeongnam Department of Environment Toxicology and Chemistry, Korea Institute of Toxicology, Jinju 52834, Republic of Korea; jdher@kitox.re.kr; 3Division of Animal Bioscience & Intergrated Biotechnology, Gyeongsang National University, Jinju 52725, Republic of Korea; hwkim@gnu.ac.kr; 4Laboratory of Developmental Biology and Genomics, BK21 PLUS Program for Creative Veterinary Science Research, Research Institute for Veterinary Science, College of Veterinary Medicine, Seoul National University, Seoul 08826, Republic of Korea; snumouse@snu.ac.kr; 5R&D Group, Kick the Hurdle, Changwon-si 51139, Republic of Korea

**Keywords:** probiotics, parabiotics, *Lactobacilus sakei* CVL-001, *Lactobacilus casei* MCL, atopic dermatitis (AD), anti-inflammation

## Abstract

As it has been revealed that the activation of human immune cells through the activity of intestinal microorganisms such as pro- and prebiotics plays a vital role, controlling the proliferation of beneficial bacteria and suppressing harmful bacteria in the intestine has become essential. The importance of probiotics, especially for skin health and the immune system, has led to the emergence of products in various forms, including probiotics, prebiotics, and parabiotics. In particular, atopic dermatitis (AD) produces hypersensitive immunosuppressive substances by promoting the differentiation and activity of immune regulatory T cells. As a result, it has been in the Th1 and Th2 immune balance through a mechanism that suppresses skin inflammation or allergic immune responses caused by bacteria. Furthermore, an immune mechanism has recently emerged that simultaneously controls the expression of IL-17 produced by Th17. Therefore, the anti-atopic effect was investigated by administering doses of anti-atopic candidate substances (*Lactobacilus sakei* CVL-001, *Lactobacilus casei* MCL, and *Lactobacilus sakei* CVL-001 *Lactobacilus casei* MCL mixed at a ratio of 4:3) in an atopy model using 2,4-dinitrochlorobenzene and observing symptom changes for 2 weeks to confirm the effect of pro-, para-, and mixed biotics on AD. First, the body weight and feed intake of the experimental animals were investigated, and total IgG and IgM were confirmed through blood biochemical tests. Afterward, histopathological staining was performed using H&E staining, Toluidine blue staining, Filaggrin staining, and CD8 antibody staining. In the treatment group, the hyperproliferation of the epidermal layer, the inflammatory cell infiltration of the dermal layer, the expression of CD8, the expression of filaggrin, and the secretion of mast cells were confirmed to be significantly reduced. Lastly, small intestine villi were observed through a scanning microscope, and scoring evaluation was performed through skin damage. Through these results, it was confirmed that AD was reduced when treated with pro-, para-, and mixed biotics containing probiotics and parabiotics.

## 1. Introduction

Atopic dermatitis (AD) is a chronic skin condition characterized by severe itching and recurring skin inflammation, and most patients experience relief over time [1,2]. AD typically affects children, but adults can also develop the disease. The symptoms of this condition may manifest differently in each individual, with some presenting with a rash, itching, dryness, swelling, or the formation of hard tissue in the affected area [3]. The pathogenesis of AD is believed to be influenced by a multitude of factors, including genetic, environmental influences, immunological abnormalities, and alterations in skin barrier function [4]. Genetic predisposition may be influenced by a family history of atopic disease. Some studies have demonstrated that variants in the filaggrin (FLG) gene represent the most significant risk factor for AD in terms of impaired skin barrier function [5,6]. The intricate interplay of the immunologic system in AD is characterized by the involvement of various subtypes of T-helper cells (Th). The Th1 subset of the T helper cells is responsible for regulating the immune response to intracellular infectious agents and secreting interferon-γ. Additionally, the Th2 subset is involved in allergic reactions and secretes cytokines such as IL-4, IL-5, and IL-13. These two T helper cells, Th1 and Th2, play an important role in AD [7,8]. Furthermore, Th17 cells secrete IL-17 in response to bacteria in order to regulate the inflammatory response. This has been demonstrated to be highly correlated with microbial imbalances [9,10,11].

While numerous factors contribute to AD, the relationship between the gut microbiome and AD represents a significant topic of recent research. In other words, researchers have reported that an imbalance in the gut microbiome can affect AD symptoms [11,12]. The gut microbiome plays an essential role in numerous bodily functions. It is responsible for regulating immune function and inflammatory responses, controlling neurotransmitters and nutrient absorption, among other things. A considerable body of research indicates that an imbalance in the gut microbiome may contribute to the pathophysiology of AD through dysregulating the inflammatory response on the gut surface [13,14]. The “gut–skin axis” hypothesis, which describes the interaction between the gut and skin, suggests that probiotics may contribute to the individualization of the gut environment and balance the immune response, making probiotics a potential therapeutic candidate for AD [11].

Probiotics are biological components that constitute the gut microbiome. The most common probiotics are *Lactobacillus* and *Bifidobacterium* [15]. Probiotics have been demonstrated to facilitate the equilibrium of the gut microbiome, which in turn enhances the functionality of the immune system by impeding the proliferation of bacteria and stimulating the activation of immune cells. Furthermore, the recent emergence of *Akkermansia muciniphila*, *Faecalibacterium*, and *Bacteroides*, collectively referred to as the “new generation of probiotics”, represents a significant advancement in the field of probiotic research [16,17].

Prebiotics represent a class of dietary components that have been demonstrated to promote the growth and activity of gut microbes, including inulin, fructooligosaccharides (FOS), and galactooligosaccharides (GOS) [18]. Parabiotics represent a distinctive category between prebiotics and probiotics. Parabiotics are inactivated strains of human beneficial bacteria that are not active in the body, rendering them safer to consume than traditional probiotics. Furthermore, they can be consumed in high concentrations and are not affected by acidic environments [19,20]. More recently, metabiotics, a combination of lactic acid bacteria metabolites and probiotics, are a new generation of bacteria that contain metabolites produced through the fermentation process, structural components of lactic acid bacteria, and immune-stimulating components, rather than simply inhibiting live bacteria like parabiotics [21]. These metabiotics can help balance the dysbiosis in the gut by keeping *Lactobacillus* alive while providing the beneficial metabolites they produce. Metabiotics are being applied in a variety of industries due to their potential immunomodulation, anticancer, and antioxidant effect [20,21].

*Lactobacillus sakei* is a Gram-positive anaerobic bacterium that is commonly found in a variety of fermented foods, including meat. The substances produced by this microorganism have been shown to inhibit the growth of harmful bacteria. *Lactobacillus sakei* CVL-001, isolated from cabbage kimchi, has been demonstrated to reduce body fat mass, plasma triglycerides, and low-density lipoprotein (LDL) [22,23,24]. In addition, *Lactobacillus sakei* recovered colonic inflammation caused by dextran sulfate through the regulation of intestinal microorganisms [25]. *Lactobacillus sakei* WIKIM30 has been demonstrated to induce regulatory T-cells and modulate gut microbiota balance in mice, thereby confirming its potential to treat AD [26].

*Lactobacillus casei* is another Gram-positive anaerobic bacterium with a wide pH and survival temperature range that is beneficial in many applications, including cheesemaking and dairy fermentation. [27]. *Lactobacillus casei* CRL 431 induces intestinal mucosal immune activation via innate immunity [28]. In addition, the *Lactobacillus casei* Sirota strain has been shown to have immune-boosting effects and to reduce abdominal stress by balancing the gut microbiome [29,30]. *Lactobacillus casei* MCL strain investigated for irritable bowel disease showed a reduced expression of pro-inflammatory cytokines such as various IL-1β, TNF-α, COX2 and INOS, and upregulated anti-inflammatory cytokines [31]. These two strains, along with the various studies to date, have commercial value and could potentially address atopic inflammatory diseases through their anti-inflammatory effects.

In this study, 2,4-dinitrochlorobenzene (2,4-DNCB) was used to induce AD in BALB/c mice. *Lactobacillus sakei* CVL-001 and *Lactobacillus casei* MCL were then administered for 2 weeks to investigate their anti-atopic effects. At the same time, we investigated the effect of a 4:3 mixture of pro- and parabiotics (*Lactobacillus sakei* CVL-001 and *Lactobacillus casei* MCL, 4:3) on AD. While previous studies have applied each strain to specific conditions, this study will provide important data on not only the effectiveness of each strain but also the mixtures of strains in treating AD.

## 2. Material and Methods

### 2.1. Strain and Atopic Agents

The two probiotic and parabiotic strains *Lactobacillus sakei* CVL-001 (KIT Code No. T0379) and *Lactobacillus casei* MCL (KIT Code No. T0380) were used as the test substances and were obtained from the Korea Institute Toxicology, Gyeongsangnamdo Jinju, Republic of Korea; the excipients were PBS. The atopic agent was 2,4-Dinitrochlorobenzene and the excipients were acetone and olive oil (3:1) (1% and 0.4%). Approximately 24 h prior to the induction of atopy, dorsal hairs were removed and sensitized by the application of 1% DNCB (2,4-dinitrochlorobenzene) (Sigma Aldrich, St. Louis, MO, USA) in a volume of 100 µL, administered once daily for three consecutive days. This was followed four days later by the administration of five applications of 0.4% DNCB (2,4-dinitrochlorobenzene) in a volume of 100 µL, administered once daily for three consecutive days, with the objective of inducing AD. The test substance was prepared on the day of dosing and administered orally (0.2 mg/head, 10 mL/kg).

### 2.2. Animals

The animals (5 animals in each group) were obtained at approximately six weeks of age and were of the specific pathogen-free (SPF) BALB/C mice. At the time of initiation of treatment, the animals were approximately seven weeks of age and weighed approximately 20% of the average total body weight. The mice were sourced from Koatek Co. (Pyeongtaek, Republic of Korea). All animals were acclimated to the laboratory environment for a minimum of five days. Following this period of acclimation, the animals were separated into groups based on their body weight, which was measured prior to the induction of atopy. The animals were housed in a poly cage (145 W × 275 D × 140 H mm) with a maximum of five animals during the acclimatization period. The housing environment consisted of wire cages (145 W × 275 D × 140 H mm) with one animal per cage during the administration and recovery period, and was subject to the following parameters: Temperature (20–27 °C), Humidity (40–60%), Lighting (12 h light/12 h dark cycle), Illumination (150–300 Lux), Ventilation (10–20 times/h). All procedures were performed in accordance with the protocols of the Institutional Animal Care and Use Committee (IACUC) of the Korea Institute of Toxicology Gyeongnam Environmental Toxicology Center. (Approval Code: B23010, Approval Date: 16 August 2023).

### 2.3. Blood Biochemistry Analysis

Approximately 1 mL of blood was collected from the posterior vena cava and allowed to stand at room temperature for approximately 90 min before centrifugation at 300 rpm for 10 min. This procedure was undertaken in order to separate the serum using a Mouse IgG (Immunoglobulin G) ELISA Kit (Elabscience Biotechnology Co., Ltd., Houston, TX, USA, Cat No. E-EL-M0692) and a Mouse IgM (Competitive EIA) ELISA Kit (LS Bio, Cat No. LS-F28569) which were then utilized to quantify the levels of IgG and IgM.

### 2.4. Measuring Skin Damage and Scratching Behavior

The mice were evaluated via clinical visual assessment after the induction of atopy. The evaluation items were erythema, itching, dryness, edema, crusting, and lichenification, and each item was scored as no symptoms (0), mild (1), moderate (2), or severe (3), and the skin damage was calculated by summing the scores of the six items. The scores for the six items were then summed to calculate the extent of the skin damage. The number of scratches was repeatedly observed for five minutes and recorded for all animals. All scoring was conducted via a blinded observation.

### 2.5. Histopathology and Microscopic Confirmation Using Skin Staining

The skin was fixed in 10% neutral formalin solution, processed using a tissue processor (Thermo Fisher Scientific, Inc., Waltham, MA, USA), and embedded in paraffin to prepare blocks. Each paraffin block was sectioned at 4 μm with a microtome. The H&E staining of the tissue slides was performed using Dako coverstainer (Agilent, Santa Clara, CA, USA). Tissues were deparaffinized using Dako coverstainers and washed and embedded in a 1% toluidine blue solution (Scytek, West Logan, UT, USA) diluted in distilled water for 5 s. The deparaffinization of the tissue was carried out using Dako coverstainers, and the tissue was reacted in a 3% hydrogen peroxide solution diluted in methanol for 30 min. Next, antigen retrieval was performed using Dako RT link. Subsequently, tissues that were to be immunostained for CD8 were blocked by reacting in 3% BSA (GenDEPOT, Katy, TX, USA) for 30 min. After that, 1 h reaction with anti-CD8 antibody diluted 1:1000 in Dako TBST buffer followed by a 30 min reaction with HRP-labeled polymer anti-rabbit secondary antibody. Likewise, there was a 1 h reaction with anti-filaggrin antibody diluted 1:4000 in Dako TBST buffer followed by 30 min reaction with HRP-labeled polymer anti-rabbit secondary antibody. All tissue slides were then stained with Dako Mayer’s hematoxylin after 30 s of DAB chromogenic reaction and 5 s of Dako Mayer’s hematoxylin staining and embedded using Dako coverstainers. The tissue results were then observed at ×200 and ×400 magnification. Histopathologic scores were evaluated by the thickness of the epidermal layer or the number of infiltrating cells.

### 2.6. Statistical Analysis

The mean and ±SEM were used to express the data. Prism software (GraphPad Inc., version 9.3.1, San Diego, CA, USA) was used for data analysis. SPSS version 12.0 was used for statistical examination (SPSS Inc., Chicago, IL, USA). One-way factorial analysis of variance (ANOVA) was used to determine whether the groups differed significantly from each other. Duncan’s multiple range and Student’s test at *p* < 0.05 level were used after the one-way analysis of variance. (# *p* < 0.05, ## *p* < 0.01, ### *p* < 0.001 vs. VC; and * *p* < 0.05, ** *p* < 0.01, *** *p* < 0.001 vs. NC).

## 3. Results

### 3.1. Body Weight

Feed intake was measured before the start of treatment and once a week after the start of treatment. On the day of measurement, a fixed amount of food was fed per cage, and the remaining amount was measured the next day to calculate the daily intake, which was calculated as the average daily intake per individual (g/rat/day) (Table 1). In all groups, there was no significant change in body weight during the experimental period.

### 3.2. Comparison of IgG and IgM Using Animal Serum

Table 2 presents a comparison of IgG and IgM in the serum of experimental animals in the normal, atopy-induced, and experimental groups. The atopy-induced and test groups exhibited significantly elevated levels of igG but not igM compared to the control group.

### 3.3. Anti-Atopic Effects of Pro- and Parabiotics Using H&E Staining

The effects of the pro- and parabiotics on AD were confirmed using hematoxylin and eosin (H&E) staining (Figure 1A). The atopic induction negative control group (NC) exhibited epidermal layer hyperplasia, folliculitis, and multiple inflammatory cell infiltrates in the dermis in comparison to the vehicle control group (VC). The atopy score for each group was expressed as a histopathologic score [32,33] (Figure 1B). Figure 1B shows that the *Lactobacillus sakei* T1 treatment exhibited a reduction in the total score of approximately 10%, which was not statistically significant. On the other hand, the *Lactobacilus casei* T2 treatment group had significantly lower histopathological scores than the NC. Treatment with *Lactobacillus sakei* T1 and *Lactobacillus casei* T2 reduced atopy, with the *Lactobacillus Sakei–Lactobacillus casei* (4:3) T3 group reducing the score by more than 70%.

### 3.4. Anti-Atopic Effects of Pro- and Parabiotics via CD8 Antibody and Filaggrin Staining

CD8 is a protein expressed on T lymphocytes, and an increase in CD8 indicates an increase in T lymphocytes and inflammation [34]. In Figure 2A, no expression of CD8 was observed in the vehicle control group (VC), whereas in the atopy-induced negative group (NC), some CD8-expressing cells were observed in the epidermal epithelium and dermal layer. In the *Lactobacillus sakei* T1, *Lactobacillus casei* T2, and *Lactobacillus sakei–Lactobacillus casei* (4:3) T3 treatment groups, somewhat reduced CD8-expressing cells were observed compared to the atopy-inducing group, and especially in the T3 treatment group, the expression of CD8 was not observed as in the normal group (Figure 2A).

In the atopy-induced group, we observed that filaggrin expression was increased in the epidermis layer, and the thickness of the epithelium was increased. Filaggrin is associated with an increased permeability of the epidermis. In the atopic induction group, filaggrin was strongly observed throughout the hyperproliferated epidermal epithelium. The *Lactobacillus sakai* T1, *Lactobacillus casei* T2, and *Lactobacillus sakai*–*Lactobacillus casei* (4:3)

T3 treatments had less expression in the atopy-induced group compared to the NC, suggesting that probiotic and parabiotic treatment can suppress atopy (Figure 2B) [35,36].

### 3.5. Anti-Atopic Effects of Pro- and Parabiotics via Toluidine Blue Staining

Mast cells in skin tissue secrete substances that regulate various anti-inflammatory factors when inflammation occurs [37]. As a result, inflammatory diseases such as AD have an increased number of mast cells to suppress inflammation [38]. In Figure 3A, mast cells were significantly increased in the atopy-induced negative group (NC) compared to the vehicle control group (VC). *Lactobacillus sakei* T1 was not significantly different from the control group. *Lactobacilus Casei* T2 treatment showed a significant decrease in mast cells compared to the control group. *Lactobacillus sakei–Lactobacillus casei* (4:3) T3 group reduced mast cells by about 60% compared to the control group, confirming that inflammatory secretion was reduced (Figure 3B).

### 3.6. Skin Damage and Scores by Date

According to the visual assessment method, the DNCB-treated NC in Figure 4 maintained skin damage such as erythema, itching, dryness, edema, crusting, and lichenification until day 15. Figure 4 shows the sum of the scores for each parameter. Nevertheless, the group treated with pro-, para-, and mixed biotics exhibited a reduction in atopic inflammation and a significant reduction in scratching behavior (Figure 4). It was concluded that probiotics, parabiotics, and mixed biotics contribute and correlate to the recovery of atopic skin and the suppression of atopy in experimental animals with atopy.

## 4. Discussion

The pathogenesis and mechanisms of AD are highly complex and involve a variety of immune-related factors. In acute lesions, Th2 and Th22 activation predominates, with IL-4 and IL13 as the main cytokines. In chronic lesions, Th1 activation predominates, and cytokines such as IFN-γ are responsible [39]. Th17 is simultaneously activated in both acute and chronic atopic lesions, and is characterized by IL-17 production [40,41].

The gut microbiome has been consistently implicated in these atopic lesions [42,43]. On this basis, gut microbiota abnormalities are associated with skin disease, which may lead to new treatment and prevention strategies for AD. [44]. In a clinical study, half of the children of pregnant women at high risk for AD who received *Lactobacillus rhamnosus* GG (*L. rhamnosus* GG) were free of AD by the age of 2 years and had a stable form of the disease by the age of 4 years. Babies treated with *L. rhamnosus* GG or the *Bifidobacterium lactis* (*B. lactis*) Bb-12 strain experienced lower rates of AD [45]. Another study investigated the interaction of *Lactobacillus rhamnosus* GG (LGG) with the skin and gut microbiome and humoral immunity in infants. The percentage of IgA- and IgM-secreting cells was significantly lower between the treated and untreated groups: after 1 month, the proportion of IgA-secreting cells was 0.59 (95% CI 0.36–0.99, *p* = 0.044) and the proportion of IgM-secreting cells was 0.53 (95% CI 0.29–0.96, *p* = 0.036) [45].

In this study, *Lactobacillus sakei* T1 was defined as a probiotic as a live lactic acid bacteria, and *Lactobacillus casei* T2 was defined as a parabiotic through the inactivation of existing probiotics. By these definitions, DNCB-induced atopic mice were treated with two strains of *Lactobacillus sakei* CVL-001 and *Lactobacillus casei* MCL. The mice were divided into five groups with a vehicle control (VC) and a negative control (NC), each receiving a single strain and a *Lactobacillus sakei*:*Lactobacillus casei* (4:3) mixture. There was no change in body weight between each group of mice. This is due to the fact that we provided the same housing conditions for the upcoming experiments, setting a baseline to control for certain variables. After that, we compared the changes in igG levels in each group and found significantly increased levels in both the NC and T groups compared to VC. In particular, igG has been shown to release inflammatory substances from skin mast cells, and we confirmed the increase in igG in DNCB-induced atopy and T group by histologic study [46].

Subsequent H&E staining confirmed that the inflammatory cell infiltration in the dermis was relieved, and CD8 antibody staining confirmed that inflammation was relieved. Filaggrin staining also showed a similar histologic morphology to normal controls in the two strain and mixed strain treatment groups compared to the atopy-induced group. When inflammatory disease is triggered by atopy, the number of mast cells increases to suppress inflammation, and the number of mast cells was studied using various histologic analyses, including toluidine blue staining, which showed a decrease in mast cells in the group treated with the two strains separately and in the mixed strain group. However, it is questionable whether histamine release from mast cells is involved in the itching and scratching behavior that is most characteristic of atopic dermatitis, and therefore the cellular mechanisms of the following AD-associated cytokines such as IL-4, IL-13, and IL-31 need to be elucidated, considering that the pathogenesis of AD is promoted by neutrophils, basophils, and CD4^+^T cells [47,48]. Finally, the modified SCORAD score was applied to assess the score for AD [49]. After 2 weeks of observation at 3-day intervals from the initial exposure phase, AD symptoms improved and the frequency of the scratching behavior decreased in the pro- (*Lactobacillus sakei* CVL), para- (*Lactobacillus casei* MCL), and mixed forms of biotics compared to the atopy-induced group.

In one study, the gut microbiota *Faecalibacterium prausnitzii* and *Akkermansia muciniphila* inhibited inflammatory cell infiltration and alleviated mast cell infiltration via the histological examination of DNCB-induced AD-like mice [50]. In another study, *Lactobacillus acidophilus* (*L*. *acidophilus*) *KBL409* strain alleviated AD in mice, showing decreased skin keratinization and epidermal thickness, downregulated Th1 (interferon-γ), Th2 (IL-4, IL-5 IL-13), and Th17 (IL-17), and upregulated the expression of IL-10, an anti-inflammatory cytokine, and Foxp3, a regulator of immune system response [51]. Therefore, these same potential mechanisms may also apply to the strains in this study. Our next study will identify these cytokines and markers and determine the proportion of bacteria that actually colonize the small intestinal villi to see if they modulate gut immunity.

Furthermore, research should include the design of human clinical trials in future to evaluate the safety and efficacy of treatments, which is critical to supporting research projects on the utility of pro-, para-, and mixed biotics in atopic dermatitis. In addition, a broader range of probiotic and combinations of strains should be investigated to determine the most effective formulations. Furthermore, conducting comprehensive studies of the gut microbiome before and after treatment will improve our understanding of how pro-, para-, and mixed biotics affect the composition of the microbiome and how these changes relate to the relief of atopic dermatitis symptoms. These strategies will provide a basic foundation for the clinical use of pro-, para- and mixed biotics in the management of this disease.

## 5. Conclusions

In conclusion, the administration of *Lactobacillus sakei CVL-001*, *Lactobacillus casei MCL*, and a *Lactobacillus sakei–Lactobacillus casei* (4:3) mixture indirectly reduced AD-associated inflammation and affected the skin barrier. These results suggest that both strains, as well as the mixture, may modulate the gut microbiota to alleviate AD, and can be further developed in future studies that profile each strain and the gut microbiota as a whole at the cellular level. Additionally, if clinical studies in humans show similar benefits, it could be a valuable treatment option. 

## Figures and Tables

**Figure 1 nutrients-16-02903-f001:**
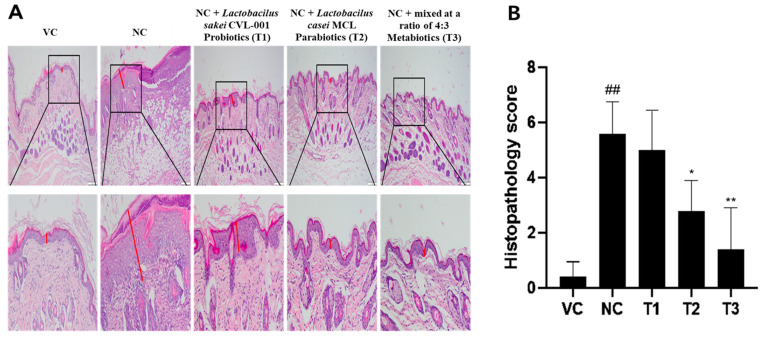
Histologic features of DNCB-induced AD-like skin damage were evaluated using H&E staining. (**A**) H&E staining of experimental animal skin for anti-atopic effects of pro-, para-, and mixed biotics (H&E × 200). (**B**) Histopathologic score in AD-like skin lesions. H&E staining showed that pro-, para-, and mixed biotics all alleviated DNCB-induced inflammation in the epidermal layer, with a histologically significant reduction in the T3 group. The red line in the figure is the relative width of the tissue stain. The data represent the mean ± SD of three independent experiments. (## *p* < 0.01 vs. VC, * *p* < 0.05 vs. NC, ** *p* < 0.01 vs. NC).

**Figure 2 nutrients-16-02903-f002:**
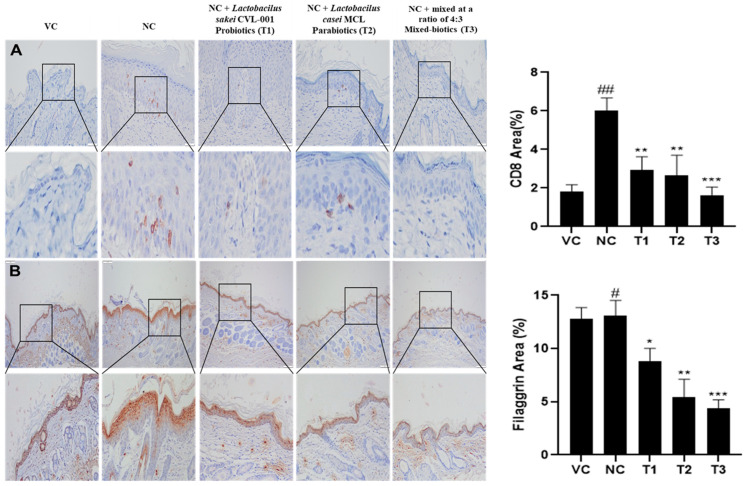
Histologic features of CD8 antibody staining and Filaggrin staining of AD-like skin lesions with DNCB. (**A**) CD8 antibody staining of experimental animal skin for anti-atopic effects of pro-, para-, and mixed biotics. Increased expression of CD8 in the epidermal and dermal layers was observed in the DNCB-induced NC, but relatively decreased CD8 expression was observed in the T1, T2, and T3 groups. Similarly, (**B**) Filaggrin staining in AD showed an increase in the thickness of the epithelial layer in the DNCB-induced NC, but a decrease in T1, T2, and T3. The data represent the mean ± SD of three independent experiments. (# *p* < 0.05 vs. VC, ## *p* < 0.01 vs. VC, * *p* < 0.05 vs. NC, ** *p* < 0.01 vs. NC, *** *p* < 0.001 vs. NC).

**Figure 3 nutrients-16-02903-f003:**
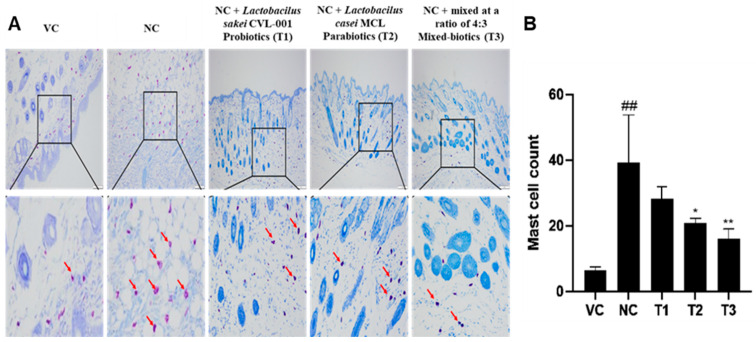
Toluidine blue staining histologic features of AD-like skin lesions with DNCB. (**A**) Toluidine blue staining of experimental animal skin for anti-atopic effects of pro-, para-, and mixed biotics. (**B**) Mast cell count in AD-like skin lesions. Mast cells increased dramatically in the DNCB-induced group, but decreased in the T1, T2, and T3 groups, with a significant decrease in the T3 group. The red arrow is the estimated number of mast cells. Data represent the mean ± SD of three independent experiments. (## *p* < 0.01 vs. VC, * *p* < 0.05 vs. NC, ** *p* < 0.01 vs. NC).

**Figure 4 nutrients-16-02903-f004:**
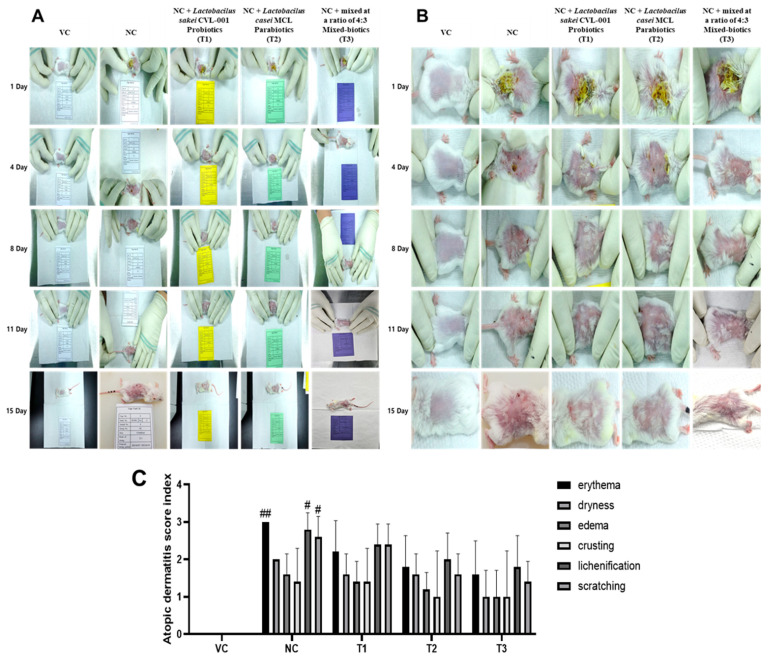
(**A**) Visual assessment and average atopic score for each symptom between days 1–15 of atopy-induced skin damage with DNCB and scratching behavior scoring results. (**B**) Enlarged the condition of atopic dermatitis in each group. (**C**) Each item is scored as no symptoms (0), mild (1), moderate (2), or severe (3). Means with different superscripts in the same row are significantly different at *p* < 0.05 via Duncan’s multiple range tests. The data represent the mean ± SD of three independent experiments. (# *p* < 0.05, ## *p* < 0.01 vs. VC).

**Table 1 nutrients-16-02903-t001:** Weekly weight changes in laboratory animals.

	Day		
	1	8	15
Vehicle Control group (VC)	24.39 ± 0.61 ^NS^	24.70 ± 0.74 ^NS^	25.18 ± 0.73 ^NS^
Atopy-induced Negative Control group (NC)	24.71 ± 1.82 ^NS^	25.06 ± 1.79 ^NS^	25.66 ± 1.31 ^NS^
*Lactobacillus sakei* T1	24.58 ± 1.35 ^NS^	24.95 ± 1.03 ^NS^	25.42 ± 1.40 ^NS^
*Lactobacillus casei* T2	24.00 ± 0.83 ^NS^	25.00 ± 1.43 ^NS^	25.65 ± 0.84 ^NS^
*Lactobacillus sakei–Lactobacillus casei* (4:3) T3	22.54 ± 1.30 ^NS^	23.22 ± 0.97 ^NS^	23.84 ± 0.87 ^NS^

All values are mean ± SD (*n* = 5). Means with different superscripts in the same column are not significantly (^NS^) different at *p* < 0.05 by Duncan’s multiple range tests.

**Table 2 nutrients-16-02903-t002:** Comparison of IgG and IgM through animal serum.

	IgG (mg/dL)	IgM (mg/dL)
Vehicle Control group (VC)	10.00 ± 2.70 ^A^	19.96 ± 2.74 ^A^
Atopy-induced Negative Control group (NC)	18.12 ± 2.16 ^BC^	21.32 ± 2.32 ^AB^
*Lactobacillus sakei* T1	16.12 ± 1.36 ^B^	22.38 ± 3.40 ^AB^
*Lactobacillus casei* T2	19.30 ± 1.97 ^BC^	27.38 ± 5.64 ^B^
*Lactobacillus sakei–Lactobacillus casei* (4:3) T3	21.17 ± 3.84 ^C^	26.10 ± 0.79 ^AB^

All values are mean ± SD (n = 5). ^A–C^ Means with different superscripts in the same column are significantly different at *p* < 0.05 by Duncan’s multiple range tests.

## Data Availability

Samples are not available from the authors due to privacy.

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
