# Peer review of "The Skin Histopathology of Pro- and Parabiotics in a Mouse Model of Atopic Dermatitis"

_nutrients, 2024, doi:10.3390/nu16172903_

Round 1
Reviewer 1 Report
Comments and Suggestions for Authors
This manuscript investigates the impact of meta-biotics, an emerging combination of probiotics and parabiotics, on atopic dermatitis (AD) using a mouse model. The study explores how specific Lactobacillus strains, administered individually or in combination, influence key immunological and histopathological markers associated with AD. The findings suggest that meta-biotics can reduce inflammation and enhance skin barrier function, offering a promising new approach for AD management. The strength of this study lies in its innovative application of meta-biotics, aligning with the growing interest in the gut-skin axis—a concept linking gut microbiota to skin health. The research employs a comprehensive methodology, utilizing various histopathological techniques to thoroughly evaluate the treatment's effects. The significant reduction in inflammatory markers and the improvement in skin barrier function underscore the potential of meta-biotics as a novel therapeutic option for AD.
However, the study has limitations. It is based on a single animal model, which may not fully capture the complexities of human AD, limiting the generalizability of the findings. The short duration of the study (two weeks) restricts insights into the long-term effects and potential side effects of the treatment. Additionally, the small sample size (5 mice per group) raises questions about the robustness of the results, and the absence of detailed mechanistic insights leaves open questions about how meta-biotics exert their effects at a molecular level.
The use of probiotics in dermatological conditions remains controversial, with clinical studies showing mixed results. While some research highlights their benefits, others show limited or no effects. This manuscript contributes to this ongoing discussion by providing experimental evidence of the potential advantages of meta-biotics in AD. As interest in the gut-skin axis grows, this study, despite its limitations, advances our understanding and suggests new avenues for therapeutic strategies in dermatology. Future research should focus on clinical trials in humans and a deeper exploration of the molecular mechanisms at play.
While the manuscript presents a novel and valuable contribution to the field, we recommend a few minor revisions to enhance its clarity and impact. First, we suggest clearly defining the term "meta-biotics" and explaining the rationale behind choosing this approach over more traditional methods. Additionally, expanding the limitations section to include specific recommendations for future research, such as the need for clinical trials in humans or a deeper analysis of molecular mechanisms, would strengthen the discussion. We also recommend briefly discussing the potential clinical implications of these findings, even if preliminary, to broaden the article's relevance to a wider audience. Lastly, if available, referencing preliminary human studies or comparative studies could further support the conclusions and reinforce the validity of the findings.
Author Response
For reviewer 1
Thank you for your attentive comments.
This manuscript investigates the impact of meta-biotics, an emerging combination of probiotics and parabiotics, on atopic dermatitis (AD) using a mouse model. The study explores how specific Lactobacillus strains, administered individually or in combination, influence key immunological and histopathological markers associated with AD. The findings suggest that meta-biotics can reduce inflammation and enhance skin barrier function, offering a promising new approach for AD management. The strength of this study lies in its innovative application of meta-biotics, aligning with the growing interest in the gut-skin axis—a concept linking gut microbiota to skin health. The research employs a comprehensive methodology, utilizing various histopathological techniques to thoroughly evaluate the treatment's effects. The significant reduction in inflammatory markers and the improvement in skin barrier function underscore the potential of meta-biotics as a novel therapeutic option for AD.
However, the study has limitations. It is based on a single animal model, which may not fully capture the complexities of human AD, limiting the generalizability of the findings. The short duration of the study (two weeks) restricts insights into the long-term effects and potential side effects of the treatment. Additionally, the small sample size (5 mice per group) raises questions about the robustness of the results, and the absence of detailed mechanistic insights leaves open questions about how meta-biotics exert their effects at a molecular level.
The use of probiotics in dermatological conditions remains controversial, with clinical studies showing mixed results. While some research highlights their benefits, others show limited or no effects. This manuscript contributes to this ongoing discussion by providing experimental evidence of the potential advantages of meta-biotics in AD. As interest in the gut-skin axis grows, this study, despite its limitations, advances our understanding and suggests new avenues for therapeutic strategies in dermatology. Future research should focus on clinical trials in humans and a deeper exploration of the molecular mechanisms at play.
While the manuscript presents a novel and valuable contribution to the field, we recommend a few minor revisions to enhance its clarity and impact. First, we suggest clearly defining the term "meta-biotics" and explaining the rationale behind choosing this approach over more traditional methods. Additionally, expanding the limitations section to include specific recommendations for future research, such as the need for clinical trials in humans or a deeper analysis of molecular mechanisms, would strengthen the discussion. We also recommend briefly discussing the potential clinical implications of these findings, even if preliminary, to broaden the article's relevance to a wider audience. Lastly, if available, referencing preliminary human studies or comparative studies could further support the conclusions and reinforce the validity of the findings.
Answer: Thank you for the reviewer's advice.
- As a first modification to your suggestion, we have reoriented the overall direction to pro-, para-, and mixed-biotics rather than meta-biotic , as there are too many gaps in our research to label the word “meta-biotics ”. I've only written a brief description of "meta-biotics" and will reserve the term meta-biotics for future strain studies.
- For the second recommendation, we added the need for clinical trials, molecular mechanisms, and future research in discussion
“Furthermore, research should include the design of human clinical trials in future to evaluate the safety and efficacy of treatments, which is critical to supporting research projects on the utility of pro, para, mixed biotics in atopic dermatitis.” (line 349-351)
“Our next study will identify these cytokines and markers and determine the proportion of bacteria that actually colonize the small intestinal villi to see if they modulate gut immunity.” (line346-348)
- Finally, we added clinical study references and preliminary human studies to further support our
“In a clinical study study, half of the children of pregnant women at high risk for AD who received Lactobacillus rhamnosus GG (L. rhamnosus GG) were free of AD by age 2 years and had stable disease by age 4 years. Babies treated with L. rhamnosus GG or the Bifidobacterium lactis (B. lactis) Bb-12 strain experienced lower rates of AD [48]. Another study investigated the interaction of Lactobacillus rhamnosus GG (LGG) with the skin and gut microbiome and humoral immunity in infants. The percentage of IgA- and IgM-secreting cells was significantly lower between the treated and untreated groups: after 1 month, the proportion of IgA-secreting cells was 0.59 (95% CI 0.36-0.99, p = 0.044) and the proportion of IgM-secreting cells was 0.53 (95% CI 0.29-0.96, p = 0.036).” (line 300-309)
“These results suggest that both strains, as well as the mixture, may modulate the gut microbiota to alleviate AD, and can be further developed in future studies that profile each strain and the gut microbiota as a whole at the cellular level. Additionally, if clinical studies in humans show similar benefits, it could be a valuable treatment option.” (line 361-365)

Reviewer 2 Report
Comments and Suggestions for Authors
This manuscript show the effects of oral administration of anti-atopic candidate substances (Lactobacilus sakei CVL-001, 29 Lactobacilus casei MCL and Lactobacilus sakei CVL-001 Lactobacilus casei MCL mixed at a ratio of 4:3) on murine model of atopic dermatitis. Though the study itself is interesting and important, there are so many issues to be solved or additional data required for publication.
Methods
2.1 Please show the number of bacteria per mL of PBS.
2.3 Why IgE level was not evaluated should be described. Hyper IgE is typical for AD.
2.4 scoring erythema, itching, dryness, edema, crushing, and tingling score: how did the authors scored itching and tingling? These are sensation and cannot be evaluated with just seeing the mice. Did the authors count scratching behaviors? How did the authors evaluate the score of tingling?
2.5 Why only CD8+ cells were counted without CD4+ cells?
Results
Line 201 Please define, C, NC, T1,T2, and T3 in Table 1.
Line 205 what is TigG or TigM? Please explain them, meaning serum IgG or IgM concentration? The serum IgE levels in each condition should be examined and the results should be shown.
Line 213 In this context, pro-, para-, and meta-biotics appear L sakei, L casei, and mixture, respectively. That is different from the definition of pro-, para-, and meta-biotics in Introduction. Please explain why the authors use such terminology for these conditions.
Figure 1 Please describe how the histopathology score is defined in Methods section. The thickness of epidermal layer, infiltrating cell number etc must be included for the histopathology score. Please add the clinical score in addition to histopathology score. Though the authors define the clinical score in Methods section, they did not at all show the results of clinical scores.
Line 245-7: The authors mentioned epidermal permeability. If they wanted to evaluate that, the authors should examine TEWL in VC, NC, T1,T2, and T3.
Figure 2 Please count CD8+ cells in high power field in each section and show the results in graph. Similarly The authors should quantify the staining intensity of filaggrin in each section.
Figure 4. Please show the figures for scratching count per hour in each condition. Please show the figures for erythema, itching, dryness, edema, crushing, and tingling scores, and total scores in each condition. Only the photographs are insufficient.
Discussion
To evaluate the balance of Th1, Th2, Th17, or Treg, the authors should examine mRNA levels of cytokines, such as IL-13, IFN-g, IL-17A, IL-22, IL-31, IL-10, TGF-beta and marker molecules, Foxp3, Tbet, GATA3, RORgt in each condition to evaluate how pro-, para-, and meta-biotics affect the balances. After showing these results, the authors should discuss how pro-, para-, and meta-biotics up- or down-regulate each Th subtype. Do pro-, para-, and meta-biotics firstly alter intestinal immune conditions, and the alteration spread systemically? What kinds of metabolites produced by these microbiota manifested these effects?
Comments on the Quality of English LanguageMinor English editing is required.
Author Response
For reviewer 2
Thank you for your attentive comments.
This manuscript show the effects of oral administration of anti-atopic candidate substances (Lactobacilus sakei CVL-001, 29 Lactobacilus casei MCL and Lactobacilus sakei CVL-001 Lactobacilus casei MCL mixed at a ratio of 4:3) on murine model of atopic dermatitis. Though the study itself is interesting and important, there are so many issues to be solved or additional data required for publication.
Methods
2.1 Please show the number of bacteria per mL of PBS.
Answer: The appropriate in vivo bacterial dosing concentration used in the experiment, 106~107 CFU, was diluted in 1 ml of PBS.
2.3 Why IgE level was not evaluated should be described. Hyper IgE is typical for AD.
Answer: The IgE levels were <2.0 in all groups, so we did not see any specific differences in the results. Instead, only the results of IgG and IgM were included in the paper, as some studies have shown a relationship between IgG and IgM in atopy.
“doi:10.3390/ijms23126867”
“https://doi.org/10.1111/j.1365-2222.1981.tb02168.x”
“https://doi.org/10.1016/j.anai.2014.12.009”
IgE |
IgA |
IgD |
IgG (mg/dL) |
IgM (mg/dL) |
|
Vehicle Control group (VC) |
< 2 |
< 5 |
< 1.33 |
10.00±2.70 A |
19.96±2.74 A |
Atopy-induced Negative Control group (NC) |
< 2 |
< 5 |
< 1.33 |
18.12±2.16 BC |
21.32±2.32 AB |
Lactobacillus Sakei T1 |
< 2 |
< 5 |
< 1.33 |
16.12±1.36 B |
22.38±3.40 AB |
Lactobacillus Casei T2 |
< 2 |
< 5 |
< 1.33 |
19.30±1.97 BC |
27.38±5.64 B |
Lactobacillus Sakei:Lactobacillus Casei (4:3) T3 |
< 2 |
< 5 |
< 1.33 |
21.17±3.84 C |
26.10±0.79 AB |
2.4 scoring erythema, itching, dryness, edema, crushing, and tingling score: how did the authors scored itching and tingling? These are sensation and cannot be evaluated with just seeing the mice. Did the authors count scratching behaviors? How did the authors evaluate the score of tingling?
Answer: All assessments were performed by a blinded observer using a visual assessment method. Scratching behavior was measured by repeated scratching over a 5-minute period and assessed for lichenification, not tingling
2.5 Why only CD8+ cells were counted without CD4+ cells?
Answer: We did not identify any specific differences in igE inthis study , which precluded histologic examination ofCD4+ , which drives the immune response , in subsequent histologic studies . Instead, we prioritized CD8+ in this study because an abnormal CD8+ response of infected and damaged cells can exacerbate skin barrier and atopic dermatitis.
Results
Line 201 Please define, C, NC, T1,T2, and T3 in Table 1.
Answer: We've defined each of the following. Vehicle Control group (VC), Atopy-induced Negative Control group (NC), Lactobacillus Sakei T1, Lactobacillus Casei T2, Lactobacillus Sakei:Lactobacillus Casei (4:3) T3
Line 205 what is TigG or TigM? Please explain them, meaning serum IgG or IgM concentration? The serum IgE levels in each condition should be examined and the results should be shown.
Answer: TigG or TigM stands for Total IgG, IgM. However, we have rewritten them as IgG and IgM to reduce reader confusion. The IgM level was not added because, as mentioned earlier, it was <2.0 in all samples.
Line 213 In this context, pro-, para-, and meta-biotics appear L sakei, L casei, and mixture, respectively. That is different from the definition of pro-, para-, and meta-biotics in Introduction. Please explain why the authors use such terminology for these conditions.
Answer: Lactobacillus Sakei T1 was defined as a live lactic acid bacteria, or probiotic, and Lactobacillus Casei T2 was defined as a parabiotic through inactivation of existing probiotics. In addition, a combination of two probiotics and a parabiotic was defined as a metabiotic to provide specific metabolites to modulate the gut microbiota and alleviate symptoms of atopic dermatitis.
However, since there is no known molecular information about meta-biotics based on this study alone, we changed the overall manuscript direction to pro- and para-biotics rather than meta-biotics
Figure 1 Please describe how the histopathology score is defined in Methods section. The thickness of epidermal layer, infiltrating cell number etc must be included for the histopathology score. Please add the clinical score in addition to histopathology score. Though the authors define the clinical score in Methods section, they did not at all show the results of clinical scores.
Answer: We added what histopathology scores include to the Methods section and added clinical outcomes.
Line 245-7: The authors mentioned epidermal permeability. If they wanted to evaluate that, the authors should examine TEWL in VC, NC, T1,T2, and T3.
Answer: We have minimized the mention of epidermal permeability in the manuscript and revised it to say that we identified potential inhibition of atopy by increased filaggrin expression in the atopic group and decreased filaggrin in the treatment group via filaggrin staining.
Figure 2 Please count CD8+ cells in high power field in each section and show the results in graph. Similarly The authors should quantify the staining intensity of filaggrin in each section.
Answer: We added graph results about CD8 and Filaggrin.
Figure 4. Please show the figures for scratching count per hour in each condition. Please show the figures for erythema, itching, dryness, edema, crushing, and tingling scores, and total scores in each condition. Only the photographs are insufficient.
Answer: The score for each condition, as follows. We've graphed the average of days 1-15 against this and added to manuscript.
Discussion
To evaluate the balance of Th1, Th2, Th17, or Treg, the authors should examine mRNA levels of cytokines, such as IL-13, IFN-g, IL-17A, IL-22, IL-31, IL-10, TGF-beta and marker molecules, Foxp3, Tbet, GATA3, RORgt in each condition to evaluate how pro-, para-, and meta-biotics affect the balances. After showing these results, the authors should discuss how pro-, para-, and meta-biotics up- or down-regulate each Th subtype. Do pro-, para-, and meta-biotics firstly alter intestinal immune conditions, and the alteration spread systemically? What kinds of metabolites produced by these microbiota manifested these effects?
Answer: In this study, we have broadly confirmed the histologic studies by treating atopic mice with pro and para-biotics (“metabiotics” word was eliminated) and in the next study, we will add the results of the actual colonization of the small intestinal villi with various markers and cytokines as mentioned by the reviewer, as well as the results of the actual colonization of the small intestinal villi shown below (lines 350-353). We would appreciate the reviewer's understanding on this point.

Round 2
Reviewer 2 Report
Comments and Suggestions for Authors
Overall, the authors well addressed the issues pointed by me and appropriately revises the manuscript. Minor English editing is recommended.
Comments on the Quality of English LanguageMinor English editing is recommended.